# Predicting Risk of Antenatal Depression and Anxiety Using Multi-Layer Perceptrons and Support Vector Machines

**DOI:** 10.3390/jpm11030199

**Published:** 2021-03-12

**Authors:** Fajar Javed, Syed Omer Gilani, Seemab Latif, Asim Waris, Mohsin Jamil, Ahmed Waqas

**Affiliations:** 1Department of Biomedical Engineering, SMME, National University of Sciences & Technology (NUST), Islamabad 44000, Pakistan; fjaved.pg@smme.edu.pk (F.J.); omer@smme.nust.edu.pk (S.O.G.); asim.waris@smme.nust.edu.pk (A.W.); mjamil@mun.ca (M.J.); 2Department of Computing, SEECS, National University of Sciences & Technology (NUST), Islamabad 44000, Pakistan; seemab.latif@seecs.edu.pk; 3Department of Electrical and Computer Engineering, Faculty of Engineering and Applied Sciences, Memorial University of Newfoundland, St Johns, NL A1B 3X5, Canada; 4Institute of Population Health Sciences, University of Liverpool, Liverpool L69 3BX, UK

**Keywords:** mental disorders, multilayer perceptrons, predictive models, public healthcare, ReliefF, support vector machines

## Abstract

Perinatal depression and anxiety are defined to be the mental health problems a woman faces during pregnancy, around childbirth, and after child delivery. While this often occurs in women and affects all family members including the infant, it can easily go undetected and underdiagnosed. The prevalence rates of antenatal depression and anxiety worldwide, especially in low-income countries, are extremely high. The wide majority suffers from mild to moderate depression with the risk of leading to impaired child–mother relationship and infant health, few women end up taking their own lives. Owing to high costs and non-availability of resources, it is almost impossible to diagnose every pregnant woman for depression/anxiety whereas under-detection can have a lasting impact on mother and child’s health. This work proposes a multi-layer perceptron based neural network (MLP-NN) classifier to predict the risk of depression and anxiety in pregnant women. We trained and evaluated our proposed system on a Pakistani dataset of 500 women in their antenatal period. ReliefF was used for feature selection before classifier training. Evaluation metrics such as accuracy, sensitivity, specificity, precision, F1 score, and area under the receiver operating characteristic curve were used to evaluate the performance of the trained model. Multilayer perceptron and support vector classifier achieved an area under the receiving operating characteristic curve of 88% and 80% for antenatal depression and 85% and 77% for antenatal anxiety, respectively. The system can be used as a facilitator for screening women during their routine visits in the hospital’s gynecology and obstetrics departments.

## 1. Introduction

Major Depressive Disorder is a chief cause of disease burden among women of childbearing age across the globe [1]. The prevalence rate of antenatal anxiety is reported to be 21–25% and perinatal depression to be 11.9%, with a higher preponderance of depression in nationals of lower income countries [1,2]. It is also reported that depression rates are inclined to be higher during pregnancy as compared to first year postpartum [3] and postnatal depression, a clinical disorder [4] is often a continuation of existing antenatal depression [3]. Worldwide, an estimated 18.2%, 19.1%, and 24.6% of pregnant women experience anxiety in their first, second, and third trimesters, respectively [5].

Depression and anxiety associated with pregnancy has classically been divided into two entities according to time of onset: during pregnancy, termed as prenatal or antenatal, and post-delivery, termed as postpartum or postnatal. In research and clinical settings, symptoms of pregnancy associated anxiety and depression are either assessed as a continuum of severity scores assessed using psychometric scales such as the Edinburgh Postnatal Depression Scale or as clinician administered diagnoses based on diagnostic criteria of International Statistical Classification of Diseases and Related Health Problems (ICD) or the Diagnostic and Statistical Manual of Mental Disorders (DSM). However, the last version of the DSM (fifth version), the distinction between antenatal or postnatal episodes has been removed. These are now known as “depressive disorder with peripartum/perinatal onset”, starting anytime during pregnancy or within the four weeks following delivery. Despite being a major health concern, symptoms of depression during pregnancy—anhedonia or changes in energy, sleep, and appetite may be misinterpreted as normal experiences of pregnancy [4].

Several investigators have delineated negative health consequences of anxiety and depression during pregnancy. For instance, children unsheltered from high prenatal maternal mood entropy have been known to report higher levels of anxiety and depression symptoms [6], poor cognitive development at age 2–3 [7], and behavioral/emotional problems of the child at age 4 [8]. Untreated depression in antenatal period holds a higher risk of low birth weight in infants, pre-term birth, severe depression in the postnatal period [9], diarrheal illnesses, poor infant growth [10,11], childhood stunting [12], increased hospitalization [13], and higher cortisol levels [14,15]. Women suffering from depression and anxiety during pregnancy report a myriad of risk factors which cluster around biological, psychological, and psychosocial themes. Previous epidemiological literature has conceptualized antenatal anxiety and depression as a multifactorial disorder with its etiology rooted in a plethora of biopsychosocial causes. A recent systematic review of 97 studies by Biaggi et al. [16] provides evidence for several risk factors that predict antenatal anxiety and depression. These factors include but are not limited to lack of social support, history of abuse or domestic violence, unplanned or unwanted pregnancy, adversities in life, perceived stress, birth by cesarean section, past pregnancy complications, and loss of pregnancy. Fisher et al. [17] reported that women at a socioeconomic disadvantage were at a higher risk of developing perinatal anxiety and depression. Recent studies have also reported the underlying biological dysregulation related with the onset of these disorders. Although the evidence is still inconclusive, a myriad of biological mechanisms has been proposed, including a dysregulated hypothalamic pituitary adrenal axis activity [18]. In addition, neuroimaging studies have also shown impaired activity regions of brain for empathy, memory, and emotion regulation [19,20].

Research in Pakistan estimates a very high prevalence of antenatal depression and anxiety [21]—a major predictor of child morbidity in the region [11]. Among Pakistani women, important risk factors include poverty, high parity, uneducated husband [21], abuse [22,23], problems in marriage, history of psychiatric illnesses, postnatal depression, previous miscarriages, stillbirths, complications in pregnancy [24], illiteracy and unplanned pregnancies [25], fear of childbirth, separation from husband [26], low social support, rural background, abortion, history of harassment, c-section delivery [27], and young age [28]. The heterogeneous etiology of perinatal depression is evident from the literature.

Pakistan also has a unique psychosocial environment; predominantly patriarchal, ridden with terrorism and internally displaced communities. These psychosocial adversities place Pakistani women most at-risk of developing antenatal depression and anxiety. These maladies combined with a weakened mental health system and psychiatric workforce in the country calls for innovative solutions to address the high prevalence. Hence, it is the need of the hour to develop evidence-based tools for screening, diagnosis, treatment, and prevention of this disorder [5]. It has also been shown that prevalence measured using symptom scales is markedly higher than prevalence derived using diagnostic instruments [1] and use of screening scales can introduce recall bias, usually offer a snapshot of an individual’s mental state, are difficult to repeat, and consume a lot of time [29].

Statistical learning-based models have shown to deliver in analyzing datasets and their predictive capability has shown promising results in developing clinical decision support systems [30]. Research shows that treatment outcomes are getting better predicted using data-derived subgroups of psychiatric patients as compared to DSM/ICD diagnoses which shows potential to reformulate major psychiatric disorders [31]. Computer aided diagnostics tools which classify mental illnesses can assist clinicians in forming more dependable, impartial, and standardized decisions in less time [32]. Machine learning techniques for classification like support vector machines [33] and neural networks [34] have been in place for a long time for data classification problems.

This work aims at finding a subset of risk factors to be used as a low cost and effort tool, that can classify if a pregnant woman is at peril of developing antenatal depression or anxiety. The rationale behind finding a subset is to obtain the minimal features that are most predictive of antenatal depression and anxiety. Although it is essential to use mental health professional-led interviews to diagnose depression and anxiety according to ICD-11 and DSM-V criteria, this requires considerable effort, time, and experience on part of the clinician. A survey-based screening for disorders is prevalent in psychiatric literature. Research has shown the Hospital Anxiety and Depression scale (HADS) also used in this study to be valid and reliable for this endeavor. According to Brennan et al. [35], for major depressive disorders, with a cut point of ≥8, HADS yields a sensitivity of 0.82 (95% CI, 0.73–0.89) and a specificity of 0.74 (95% CI, 0.60–0.84) and a cut point ≥11 gives a sensitivity of 0.56 (95% CI, 0.40–0.71) and a specificity of 0.92 (95% CI, 0.79–0.97). An early prediction of the risk can help clinicians provide appropriate and timely treatment reducing the risk of the problem exuberating. This also offers reduced patient-doctor communication costs. It can further be incorporated in gynecologist and obstetric settings to keep the antenatal mental health of women in check. To the best of our knowledge, this is the first predictive modelling done for antenatal depression and anxiety.

## 2. Related Work

### 2.1. Perinatal Depression and Machine Learning

To determine the social-demographic risk factors of postpartum depression in 1447 Turkish women, a comparison between logistic regression and classification tree method was made [36]. While the classification tree method (maximal) was more articulate in detailing diagnosis, it was observed that classification trees are difficult to use practically. Logistic regression model and optimal tree showed lower sensitivity.

Tortajada et al. [37] found that in a total of 1397 Spanish women, social support, neuroticism, life events, and depressive symptoms immediately after delivery were pivotal in the prediction of post-partum depression. They presented four artificial neural network models with two feature subsets, with and without pruning attaining the highest area under the receiver operating characteristic curve of 84%. The same dataset was utilized dropping any variables that increased the cost of assessing risk and were significant according to literature on postpartum depression [38]. This study used four different classifiers namely naïve Bayes, logistic regression, support vector machines, and artificial neural networks to predict post-partum depression (PPD) after week 32 of childbirth. Their model attaining the highest AUC of 75% was logistic regression.

In another study, long short-term memory (LSTM) artificial recurrent neural network was used to screen for perinatal depression in 446 WeChat users in China. They proposed using emoticons as features. The paper reports similar results as that of Edinburgh Postpartum Depression Scale [39].

Moreira et al. [40] utilized decision trees, support vector machines, ensemble classifiers, and neural network classifiers in biomedical and socio-demographic data analysis to predict post-partum depression in Brazilian women who had suffered from hypertensive disorders during pregnancy. Results showed that ensemble classifiers were most effective in predicting risk outcomes in women.

### 2.2. Antenatal Depression in Pakistan

The Schedule for Clinical Assessment in Neuropsychiatry (SCAN) was used to diagnose a sample of 632 women with ICD-10 depressive disorder in Kahota, district of Rawalpindi, Pakistan [21]. The prevalence rate of antenatal depression was 25%. Uneducated husband, lack of a close adviser, poverty, and higher parity were concluded to be the risk factors. It was also observed that these women scored higher on Self-Reporting Questionnaire and Brief Disability Questionnaire prenatally and had a history of adverse life events in the year prior to the third trimester of pregnancy as compared to the women whose depressive disorder cleared.

In a sample of 1368 women in Hyderabad, Sindh, husband’s unemployment, unwanted pregnancy, net worth, and 10 years or more of formal education were reported as significant factors of antenatal depression and anxiety [22]. The top-level predisposing factors associated with depression and anxiety were physical, verbal, and sexual abuse.

Imran et al. [24] reported a prevalence rate of antenatal depression to be 42.7% in a sample of 213 women in Lahore, Pakistan. Depressed women were reported to have problems in marriage, with parents/in-laws’ history of domestic violence, psychiatric problems, and postnatal depression. In obstetric outcomes, previous miscarriages, stillbirths, and complications in previous pregnancies were reported as significant factors of depression.

Shah et al. [23] reported a high prevalence (48.4%) of antenatal depression in 128 pregnant women from Northern Pakistan. Depression was associated with impoverished physical health which co-related to physical abuse.

The Aga Khan University Anxiety and Depression Scale (AKUADS) was used to assess the presence of antenatal depression in 340 pregnant women in Chitral, Pakistan [25]. Using a cutoff of 13, the presence of antenatal depression was reported to be around 34%. Verbal and physical abuse, unplanned pregnancy, and illiteracy were independently related to depression.

In a sample of 506 pregnant women in Lahore, Pakistan, the strongest risk factors highlighted of antenatal depression were fear of birthing and parting of ways with husband, while history of domestic violence, narcotics abuse, lack of a support system, miscarriage, and personal history of any psychiatric illness were ruled out. [26].

Safi et al. [41] reported an 80% prevalence of antenatal depression in a sample of 300 women from Peshawar who visited Hayatabad Medical Complex for their antenatal visits. Factors identified as high risk included illiteracy, unemployment, low-income level, extended family, adverse pregnancy outcome, and fear of childbirth.

Waqas et al. [27] reported antenatal anxiety and depression in 500 pregnant women to be significantly associated with low social support, rural background, history of harassment, abortion, cesarean delivery, and unplanned pregnancies. Higher number of daughters was directly associated with women scoring higher on Hospital Anxiety and Depression Scale (HADS).

The study [42] reported a 43% prevalence of antenatal depression in an assessment of 82 women from Lahore, Pakistan. The women were in their 2nd trimester and were screened via the Edinburgh Postnatal Depression Scale (EPDS).

The prevalence of depression in expecting women attending antenatal clinics in Karachi, Pakistan was determined [28]. Among 300 women, the prevalence of antenatal depression was reported to be 81%. Risk factors included young age, parity, and living in joint families.

## 3. Methodology

In this work, we aim to classify people at risk of antenatal depression and anxiety. We used two classifiers namely multi-layer perceptron based artificial neural networks (ANNs) and support vector machines (SVMs). The key phases of the proposed methodology include acquiring the dataset, exploratory data analysis and pre-processing, predictor selection, data transformation, modeling, evaluating models, and presenting results. Figure 1 demonstrates the steps in the proposed methodology. The dataset associated with this manuscript is available as Appendix A and the code at github https://github.com/FJaved1/antenatal-depression-prediction.git (accessed on 10 March 2021).

### 3.1. Dataset

The data came from a cross sectional study that took place (February 2014–June 2014) in four teaching hospitals in Lahore, Pakistan. A total of 500 pregnant women were conveniently sampled from hospital’s obstetrics and gynecology departments when they visited for their routine prenatal care. All participants included were from low/lower-middle socio-economic status. The women were informed of the objectives of the study and a written consent was obtained from all those who agreed to take part in the study. All participants were interviewed by trained medical students enrolled in Combined Military Hospital- Lahore Medical College. Prior, these students were tutored in a 2-day workshop conducted by experienced psychologists at the Department of Psychology, CMH-LMC for interviewing. Each woman was interviewed for filling questionnaires consisting of following sections: the Hospital Anxiety and Depression Scale (HADS) and the Social Provisions Scale (SPS), and Socio-Demographics questionnaire [27]. The key clinical outcomes depression and anxiety were assessed using the Urdu translation of Hospital Anxiety & Depression Scale (HADS) [27]. Regarding cross-cultural and criterion validity, HADS has been evaluated in Pakistan [43]. It consists of 14 questions, with 7 questions each assessing depression and anxiety. Each scale is measured from 0–21 with a higher score associated with increased depression/anxiety. An Urdu translation of Social Provision Scale (SPS) was used to assess Perceived Social Support [44]. This is a 24-question scale with a Likert-type response scale, ranging from 1 (strongly disagree) to 4 (strongly agree). SPS measures the participants’ perceived reliable alliance, attachment, nurturance, guidance, reassurance of worth, and social integration with their current relations e.g., family, friends, community, and co-workers.

In addition to using validated measures for evaluation of antenatal anxiety and depression, an interviewer administered proforma was used to ask the respondents’ demographic characteristics such as age, background, and ethnicity. This was followed by several questions on obstetric history, outcome of previous pregnancies, and age and gender of previous children. Questions regarding harassment and domestic abuse were assessed using a dichotomous question. A detailed checklist was utilized to assess previous life adversities such as death of husband or parents, loss of children, and recent illnesses requiring hospitalizations.

### 3.2. Data Preprocessing

Exploratory data analysis revealed that the prevalence rate of antenatal depression and anxiety in the dataset calculated using HADS were as indicated in Table 1. The data that was acquired had some missing values indicated in Table 2. There has been ample of research on methods to fill missing values [45]. However, the variables with missing values in this dataset were such that they could be filled from information obtained from other variables of the same participant. Live births and still births were filled using logical evidence from other variables such as children total, miscarriage, and total deliveries of the participant. Maternal age was dropped from further analysis since the empty values could not be filled. The empty value of the participant in the variable adverse outcomes (during previous pregnancy) was updated according to her pregnancy history. The two empty values in Stat 4 and Stat 5 variable (questions in Social Provision Scale) were filled keeping in view that these were left unfilled by the participant. The outcome variables depression and anxiety were set at (0–7, normal) and (8–21, borderline depressed/depressed and borderline anxious/anxious) in order to decide whether the patient was suffering from depression/anxiety (positive class) or not (negative class). Variables with repeated or redundant information were removed. Variable husband death with zero variance was removed as this variable will be of no use in predicting outcome. Children total, total deliveries, highly correlated with live births, were removed as the feature selection algorithm used cannot handle multi-collinearity.

### 3.3. Model Development

Data are split in a training (80%) and a test set (20%) using stratified shuffle split such that both sets follow the prevalence rate of anxiety/depression in the original database. A feature selector was used to select the top predictors of outcome variables (depression/anxiety) from the training set. This new predictor set was then used to tune a neural network classifier and a support vector machine classifier. We used grid search hyperparameter optimization with 10-fold cross validation to select the hyperparameters of the models. Using the predictors selected from the feature selection step, the tuned models were then used to predict on the external hold-out test set. Variables used for feature selection after initial data cleaning are indicated in Table 3.

### 3.4. Predictor Selection

The aim of feature selection in this work is two-fold: first, to analyze and highlight predictive features that contribute most to the risk of depression and anxiety, and secondly, to reduce the resources required to screen for the risk of depression and anxiety. Feature selection aims at finding a small set of features from the original data space with an objective to remove the redundant and irrelevant features which otherwise would increase the computation time and effort [46]. The minimum number of best features are found by utilizing a feature selection algorithm which uses the ranking criteria method. Currently, there is no universal best feature selection method that works for all types of input data [47].

This work uses ReliefF [48], the best-known variant of Relief [49]. Relief is a univariate, only filter algorithm capable of detecting feature dependencies. Unlike wrapper and embedded methods, selected features are not induction algorithm-dependent.

Relief uses feature weights as a statistic to score a feature’s quality. It starts with randomly sampling m training instances with p features long, feature weight vector of zeros (W). In every iteration, the algorithm takes a feature vector X coming from one randomly sampled instance belonging to m and the feature vectors of the instance closest to X (by Euclidean distance) from each output class (0,1). The closest same-class sample instance is called “*near-hit*”, and the nearest different-class sample instance is called “*near-miss*”. The algorithm rewards the feature if the sample has a large distance to its nearest neighbor sample from the opposite class and a small distance from its nearest neighbor sample from the same class. The feature is regarded as good when all samples support the above rules [46].

The weights are updated as indicated in [1].
(1)$$Wi\;=\;Wi\;-\;{\left(x_{i}\;-\;nearhit_{i}\right)}^{2}\;+\;{\left(x_{i}\;-\;nearmiss_{i}\right)}^{2}$$

Scikit-Rebate [50] is used for feature selection on the training set (80%). TURF iterative feature selection wrapper [51] is used with the core ReliefF algorithm. TURF is a recursive feature elimination approach that can be used with any core Relief-based algorithm (RBA). After m iterations, feature scores are divided by m resulting in the final relevance sore. The relevant features are then selected based on scores above a certain threshold T. This threshold is determined by Chebyshev’s inequality for a confidence interval of 0.75, resulting in a threshold value of 0.1 (for m = 120). Independent variables used for modeling after feature selection are indicated in Table 4.

### 3.5. Encoding and Scaling

Machine learning algorithms require numerical input. All the data came numerically encoded which worked for binary and ordered categorical variables. The nominal categorical variables were then one-hot-encoded. One-hot-encoding creates binary vectors of the categories of a variable such that that the categories have no ordered relationship [52]. All variables except the one-hot-encoded were standardized to have a mean of zero and standard deviation of one [53,54].

### 3.6. Classification

#### 3.6.1. Support Vector Machines

A classifier that can be used for predictive modeling is support vector machine. It works by dividing the training data through a maximum margin hyperplane which can subsequently be used to classify new input data points [55]. Non-linear classifiers were later proposed which used the kernel trick for projecting the data into a higher dimension for classification not possible linearly [56].

In this work we used GridSearchCV [33] and 10-fold cross validation (CV) [57] to find the optimal SVM parameters. The parameter grid was set using multiples of 10 and is detailed in Table 5. To improve the model performance, the parameter grid was fine-tuned and updated according to the results of each run. Parameters we considered for tuning were kernel function and c. The kernel parameter decides the type of kernel function to be used in the algorithm. The parameter c is the penalty parameter for error term allowing a trade-off between higher or lower misclassification rate. A high value of c will penalize the cost of misclassification leading to a tightly fit boundary separating the training points of two classes whereas a lower c allows misclassifications. The optimal values of kernel and c were computed via GridSearchCV and later evaluated using 10-fold CV. The found values were then used to train the SVC on the training set (80%) whereas the results were assessed on the hold-out test set (20%).

#### 3.6.2. Artificial Neural Networks

The artificial neural network used in this study is a multi-layer perceptron (MLP). The MLP is represented as connected layers of nodes. The three layers in all MLP are (1) input layer, (2) output layer, and (3) one or more hidden layers. Each node in the subsequent layer takes a weighted input from all nodes of previous layer. An activation function applied to incoming weights on each node creates complex non-linear mappings between the network’s inputs and outputs. Backpropagation is used with an optimization method gradient descent to minimize a loss function. The loss function is minimized by updating the weights of the network. The network weights are updated after each iteration [58].

This work uses a feed-forward artificial neural network. During the development of the model, there are many parameters known as hyperparameters that need optimization. This optimization is needed to correctly classify instances. The number of nodes in the input layer are equal to the input features. The neural network has a logistic output. The logistic output is chosen so that model’s output could be interpreted as a probability of risk of depression and anxiety between 0 and 1. The output layer has one node.

For fine-tuning the neural network, a validation set (10%) is extracted from the training set (80%) using stratified split. Adam optimizer [59] with a learning rate of (0.1, 0.01, 0.001, 0.0001) was used to assess the performance on the validation set. Adam is an optimization algorithm that minimizes the loss function by adjusting the weights of the network. Adam leverages the advantages of Adagrad [60] and RMSprop [61]. Training a network is an iterative process where we want to minimize the loss function.

The number of hidden layers and nodes of the neural network were set and evaluated using informed hit and trial via the learning curves and cross-validation performance. To fix the number of nodes, a constructive approach was used. The constructive approach starts with a minimal network and adds additional hidden nodes. One to three hidden layers were assessed. It was observed that two hidden layers with few nodes had approximately same performance as one hidden layer with large number of nodes. It has also been shown that two hidden layers generalize better than one [62].The number of nodes in these hidden layers were first evaluated using Masters’ geometric pyramid rule [63] for one [3] and two hidden layers [4,5]. Here,
(2)$$r=\;\sqrt[{3}]{\frac{n}{m}}$$

*n* = nodes in input layer

*m* = nodes in output layer

In case of 1 layer,
(3)$$no.of\;nodes\;in\;hidden\;layer=\;\sqrt{n\times m}$$

In case of 2 layers,
(4)$$no.of\;nodes\;in\;hidden\;layer\;1=m\times r^{2}$$
(5)no.of nodes in hidden layer 2=m×r

We tried various combinations of nodes using nodes calculated from [3,4,5] as a starting point until the increase in complexity did not increase the classification score significantly. It was also observed that optimal nodes were not restricted to an explicit quantity.

Training is continued till the error on the validation set keeps dropping. Training is halted when validation error starts to increase as these are early signs of model overfitting. Various weight initializers, available in Keras, were assessed.

Overfitting in neural networks refers to the phenomenon where a neural network overlearns the training data such that it is unable to generalize on the data it has not been trained on. Hence, regularization becomes critical for neural network. This network uses ridge regression performing L2 regularization which adds a penalty term to the loss function such that weight vectors shrink at each step while the usual gradient update takes place. This network uses an L2 weight penalty of 0.01. A smaller penalty than this is not effective in reducing overfitting whereas a larger penalty increases the bias in the network.

With the selection of different hyperparameters, a 10-fold cross validation [57] is run over the entire training set (80%) to assess the performance of selected hyperparameters.

### 3.7. Evaluation Metrics

Model performance is assessed using metrics derived from the confusion matrix and area under the receiver operating characteristics. The evaluation metrics are defined below.

Confusion matrix: The confusion matrix is detailed in Table 6.

True positives are the positive cases predicted as positives. True negatives are the negative cases predicted as negatives. False positives are the negative cases predicted as positives (Type I Error). False negatives are the positive cases predicted as negatives (Type II Error).

Accuracy: accuracy is the percentage of correctly classified instances.
(6)$$Accuracy=\;\frac{TP+TN}{TP+TN+FP+FN}\times 100$$

Sensitivity: sensitivity is the percentage of true positives which get predicted as positives. Sensitivity is also known as recall or true positive rate.
(7)$$Sensitivity\;=\frac{TP}{TP+FN}\times 100$$

Specificity: specificity is the percentage of true negatives which get predicted as negatives. Specificity is also known as true negative rate.
(8)$$Specificity\;=\frac{TN}{TN+FP}\times 100$$

Precision: precision measures if percentage of predicted positives is truly positives.
(9)$$Precision\;=\frac{TP}{TP+FP}\times \;100$$

F1 Score: weighted average of *precision* and *recall*.
(10)$$F1\;=\;2\;\times \frac{precision\times recall}{precision+recall}$$

Area under the receiver operating curve: it measures the performance of classifier at various thresholds. AUC is plotted with the true positive rate on y-axis and (1—Specificity) on x-axis.

All neural network models were developed and applied using Keras (Python) [64]. SVM models and performance metrics were calculated with Scikit-learn [65]. The random seed used for splitting data in the entire work is 1. All work was executed on windows operating system, Intel Core (TM) i7-7500U CPU @ 2.70GHz, 2904 MH, 2 Core(s), with 4 Logical Processor(s).

## 4. Results and Discussion

We selected the top 3 and 5 predictors of antenatal depression and anxiety, respectively, using ReliefF. We then built a machine learning model using this set of variables. The parameters used to train the SVM are indicated in Table 7. The final MLP-based neural network used for predicting antenatal depression and antenatal anxiety had the parameters indicated in Table 8.

The trained models with these hyper parameters selected from the validation set, were then used to assess the external hold-out test set. We report the average performance with standard deviation over 30 trials in Table 9 and Table 10 and box plots of these trials in Figure 2 and Figure 3. Each trial consisted of weights being randomly initialized followed by model training and testing. We also report five repetitions of these 30 trials in Figure 4 and Figure 5 which indicate the robustness of model.

The cross-validation results of neural network in Table 9 and Table 10 indicate the mean and standard deviation of one run of 10-fold cross validation. This run indicates both data and model variance. The data was spilt using the same constant random seed. The outputs are set at default probability of 0.5. FPR is the false positive rate and FNR is the false negative rate.

In the prediction of a disease, we want to make sure that we correctly predict as many positives cases as possible as well as ensure that that the predicted positive cases are actually positives hence precision and recall are important performance measures. Multilayer perceptron and SVM achieved a sensitivity of 87% and 78% for antenatal depression, respectively. A precision of 91% was achieved by MLP and 84% by SVM. In the antenatal anxiety model, MLP (90%) scored higher precision while SVM (95%) achieved higher sensitivity. The model achieved an AUC of 88% and 85% for antenatal depression and anxiety neural network model on the hold-out test set suggesting reasonable predictive differentiation ability of the selected predictors.

In a survey of applications of neural networks in real-world scenarios, it has been shown that feed-forward neural networks have been performing better in applications to human problems [66]. This work endorses this since the multi-layer perceptron based neural network outperformed the support vector classifier in all key metrics. The neural network was also able to generalize well to the hold-out test set as indicated in Table 9 and Table 10. The support vector classifier was able to generalize only for the anxiety model and not for the depression model. One of the highlighting contributions of this work was to achieve a trimmed dataset (only 3 variables to detect the risk of antenatal depression, and 5 variables to detect the risk of antenatal anxiety) from a dataset of 36 independent variables.

Appendix A present the number of people depressed and anxious, respectively, grouped by their selected variables. Appendix A shows that people on the lower spectrum of perceived social support and their husbands being the primary decision makers of the house self-screened themselves to be more depressed. The same was observed for people with high anxiety scores as shown in Appendix A. Hence, empowerment in the houses can be an important predictor of both antenatal depression and anxiety. Similarly, women with low social support living in rural areas were found to be more prone to depression as opposed to women with high perceived social support living in urban areas as shown in Appendix A, Appendix A. This can partly be attributed to the presence of basic life facilities like health, housing, and education. Appendix A indicates that the women who used planning methods screened themselves to be less depressed than the women who did not use planning methods. Women with unplanned pregnancies grouped with no planning methods employed had the highest anxiety scores. Appendix A indicates that young adults aged between 20 and 30 were more prone to antenatal anxiety.

Mental health is reported to be fifth in line to contribute to the global burden of disease. The economic cost associated with it is estimated to go up to US 5 trillion$ by 2030 [67]. Deep learning in psychiatry can therefore assist in predicting symptoms onset or risk which can potentially open new avenues for affordable and context-based interventions at early stages in assisting and reducing this burden [32]. In this study, we analyzed the classification performance of support vector machine and multi-layer perceptron-based artificial neural network for the prediction of risk of antenatal depression and anxiety. ReliefF was used for feature selection prior to modeling. MLP-based neural networks generalized better on the hold-out test set than support vector classifier. Moreover, in the future, ANNs can be more efficient and effective in handling larger datasets which may pan out to more complex problems. Perceived social support, background, and empowerment in their houses were found to be the risk factors for antenatal depression while perceived social support, empowerment in houses, planned pregnancy, ever used planning methods, and age were risk factors found for antenatal anxiety. In a systematic review which identified women at risk of antenatal depression and anxiety [16], the most relevant factors found were lack of social or partner support, abuse, adverse events in life, stress, complications in pregnancy, and unplanned and unwanted pregnancy; three of which have been identified to be risk factors by our research as well. Since causality is of utmost importance in any interventional task in healthcare, further work is warranted to establish causal risk factors. For real-world deployment, robust methods need to be developed since machine learning methods in healthcare must exhibit the highest level of interpretability, generalizability, and robustness.

There are several limitations to this study. First, due to constraints in the availability of data, we were only able to investigate risk factors available for one city of Pakistan, i.e., Lahore. Hence, for this model to be applicable to wider population, further analysis must be done for data available for that wider population. Second, all the risk factors mentioned in the literature could not be investigated due to their absence in collected data.

## 5. Conclusions

In summation, this work proposes an artificial neural network classifier; AUC (88% for antenatal depression and 85% for antenatal anxiety) and support vector machine classifier (80% for antenatal depression and 77% for antenatal anxiety) to predict the risk in expecting mothers. Clinical decision support systems are needed and are predicted to be more effective owing to their ability to maintain a consistent quality which is distinctly lacking in low resource settings. The need for high value care with minimal cost while maintaining the standards calls for clinical decision support systems that can intelligently channel our resources to people in low resource settings. [68].

## Figures and Tables

**Figure 1 jpm-11-00199-f001:**
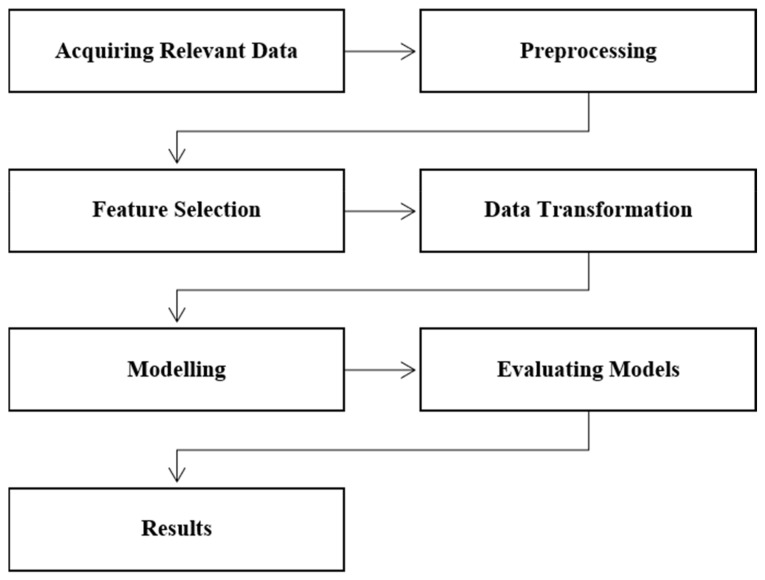
Steps in the proposed analytic methodology.

**Figure 2 jpm-11-00199-f002:**
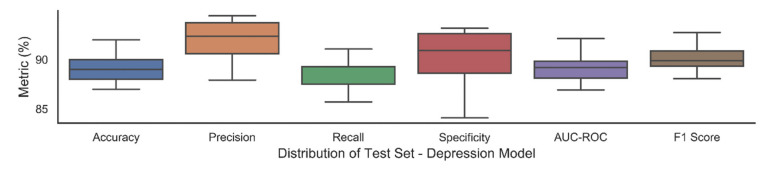
Box plots exhibiting average performance with standard deviation over 30 trials for depression.

**Figure 3 jpm-11-00199-f003:**
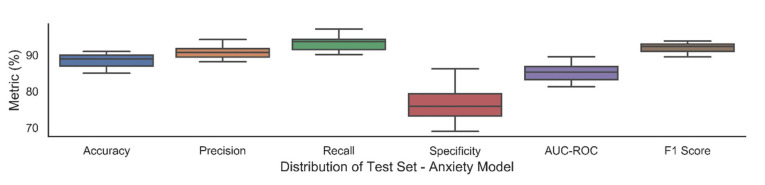
Box plots exhibiting average performance with standard deviation over 30 trials for anxiety.

**Figure 4 jpm-11-00199-f004:**
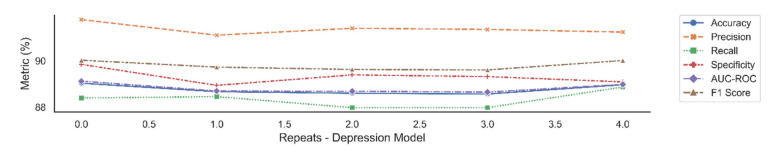
Average performance with standard deviation over 30 trials, for depression, over five repetitions.

**Figure 5 jpm-11-00199-f005:**
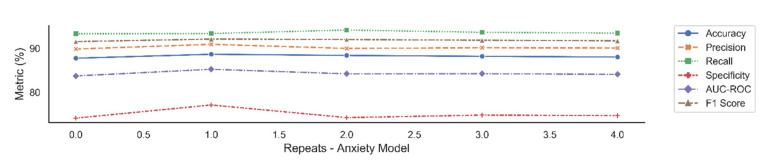
Average performance with standard deviation over 30 trials, for anxiety, over five repetitions.

**Table 1 jpm-11-00199-t001:** Prevalence rates.

Disorder	Depressed	Anxious
Positive	56.4%	71%
Negative	43.6%	29%

**Table 2 jpm-11-00199-t002:** Missing values.

Variable	# of Missing Values
Live births	27
Still births	68
Maternal age	100
Adverse outcomes	1
Stat 4	1
Stat 5	1

**Table 3 jpm-11-00199-t003:** Variables used for feature selection after pre-processing.

Social Provision Scale Questionnaire	Current Age	Ethnicity	Education	Occupation	Background
Household income	Duration marriage	Household decision maker	Fight/arguments with in-laws	Number of people living in the house	Smoking
Substance abuse	Maternal age new	Planned pregnancy	Menstrual cycle history	Ever used planning methods	Live births
Still births	Adverse outcomes during previous pregnancy	Abortion history	Past psychiatric illnesses	Psychiatric illnesses in family	Child death
Miscarriage	Parents’ death	Total male children	Relationship problems	Long illnesses	Any other past trauma
Harassment	Ever experienced domestic violence	Total spontaneous vaginal deliveries	Total episiotomies	Total c-section	Total female children

**Table 4 jpm-11-00199-t004:** Features selected using ReliefF.

Depression	Anxiety
Social support	Social support
Household decision maker	Household decision
Background	Planned pregnancy
	Ever used planning methods
	Age

**Table 5 jpm-11-00199-t005:** Parameter grid for support vector machine (SVM).

Kernel	C ^1^
Linear	0.01
Poly	0.1
RBF	1
Sigmoid	10

^1^ C _=_ regularization parameter.

**Table 6 jpm-11-00199-t006:** Confusion matrix.

		Actual Label	
		Positive [1]	Negative [0]
Predicted label	Positive [1]	True positives	False positives
	Negative [0]	False negatives	True negatives

**Table 7 jpm-11-00199-t007:** Parameters used to train SVM.

Support Vector Machine	Antenatal Depression	Antenatal Anxiety
C ^2^	1.3	1
Kernel	poly	rbf
Degree	2	Not Applicable

^2^ C = regularization parameter.

**Table 8 jpm-11-00199-t008:** Parameters used to train neural networks.

Hyperparameters	Antenatal Depression	Antenatal Anxiety
Layer	Dense Layer	Dense Layer
Topology	31-11-7-1	31-21-1
Activation function for all layers except output	RELU	RELU
Activation function for output layer	Sigmoid	Sigmoid
Epochs	90	70
Batch size	32	32
L2 weight decay	0.01	0.01
Optimizer	ADAM	ADAM
Learning rate	0.001	0.001
Loss function	Binary cross entropy	Binary cross entropy
Kernel initializer	Xavier	Xavier

**Table 9 jpm-11-00199-t009:** Results for antenatal depression neural network (NN) model.

Antenatal Depression	Evaluation Metrics	Accuracy	Sensitivity	Specificity	Precision	F1 Score	AUC-ROC	FPR	FNR
MLP-NN	CV Score	79.272	76.245	83.235	85.963	80.448	79.740	16.765	23.755
	mean(std)	(6.031)	(9.211)	(8.977)	(6.716)	(6.045)	(5.921)	(8.977)	(9.211)
	%								
	Test set	**88.600**	**87.976**	**89.394**	**91.411**	**89.628**	**88.685**	**10.606**	**12.024**
	mean(std)	**(1.281)**	**(2.110)**	**(2.835)**	**(2.023)**	**(1.165)**	**(1.338)**	**(2.835)**	**(2.110)**
	%								
Support Vector Machine	CV Score	82.500	74.271	93.098	93.422	82.268	83.685	6.902	25.729
	mean(std)	(5.701)	(10.730)	(5.333)	(5.098)	(6.986)	(5.332)	(5.333)	(10.730)
	%								
	Test set	80.0	78.6	81.8	84.6	81.4	80.2	18.1	21.4
	%								

AUC-ROC—area under the receiver operating characteristic curve, FPR—false positive rate, FNR—false negative rate. Bold values indicate best results on test set among the two classifiers used.

**Table 10 jpm-11-00199-t010:** Results for antenatal anxiety NN model.

Antenatal Anxiety		Accuracy	Sensitivity	Specificity	Precision	F1 Score	AUC-ROC	FPR	FNR
MLP-NN	CV Score mean(std) %	80.235 (5.157)	90.099 (5.865)	56.136 (13.897)	83.616 (4.503)	86.586 (3.580)	73.117 (6.950)	43.864 (13.897)	9.901 (5.865)
	Test set mean(std) %	**88.667 (1.738)**	93.380 (1.750)	**77.126 (4.315)**	**90.930 (1.566)**	**92.125 (1.213)**	**85.253 (2.305)**	**22.874 (4.315)**	6.620 (1.750)
Support Vector Machine	CV Score mean(std) %	79.250 (6.805)	88.709 (6.534)	58.431 (13.344)	83.532 (8.427)	85.637 (5.066)	73.570 (7.429)	41.569 (13.344)	11.291 (6.534)
	Test set %	85.0	**95.8**	58.6	85.0	90.0	77.1	41.3	**4.2**

AUC-ROC—area under the receiver operating characteristic curve, FPR—false positive rate, FNR—false negative rate. Bold values indicate best results on test set among the two classifiers used.

## Data Availability

The dataset associated with this study, has been provided as Appendix A.

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
