# Peer review of "Predicting Risk of Antenatal Depression and Anxiety Using Multi-Layer Perceptrons and Support Vector Machines"

_jpm, 2021, doi:10.3390/jpm11030199_

Round 1

Reviewer 1 Report

The paper explores predictors of antenatal anxiety and depression by application of machine learning technologies. Selected methods of primary assessment are appropriate given the objective and the context of the study. The machine learning analysis methods, e.g. support vector machines and neural networks allow for advanced level of data processing and delivery of sound results. The paper may benefit from additional information (250-330 words) on the underlying biological mechanisms of antenatal depression and anxiety as well as from a concise conclusion to summarize the findings and provide recommendations for clinical decision making.

Author Response

Comment 1

The paper may benefit from additional information (250-330 words) on the underlying biological mechanisms of antenatal depression and anxiety.

Response

Thank you so much for your excellent comments. Although biological mechanisms of antenatal anxiety and depression could be important components of this manuscript, the studies citing biological mechanisms are still inconclusive. Therefore, adding a lot of information on this would have deviated the reader from objectives of the study as well as reduced the awareness of this disorder as biopsychosocial entity. Therefore, we have augmented our previous paragraphs on etiology of depression to include social, biological, and psychological causes. The paragraph reads as,

“Women suffering from depression and anxiety during pregnancy report a myriad of risk factors which cluster around biological, psychological, and psychosocial themes. Previous epidemiological literature has conceptualized antenatal anxiety and depression as a multifactorial disorder with its etiology rooted in a plethora of biopsychosocial causes. A recent systematic review of 97 studies by Biaggi et al. (16) provides evidence for several risk factors that predict antenatal anxiety and depression. These factors include but are not limited to lack of social support, history of abuse or of domestic violence, un-planned or unwanted pregnancy, adversities in life, perceived stress, birth by cesarean section, past pregnancy complications and loss of pregnancy. Fisher et al. (17) reported that women at a socioeconomic disadvantage were at a higher risk of developing perinatal anxiety and depression. Recent studies have also reported the underlying biological dysregulation related with the onset of these disorders. Albeit, the evidence is still inconclusive, a myriad of biological mechanisms has been proposed including a dysregulated HPA axis activity, in addition, neuroimaging studies have also shown impaired activity regions of brain for empathy, memory and emotional regulation.” (Line 99 to 104)

Comment 2

A concise conclusion to summarize the findings and provide recommendations for clinical decision making.

Response

We have now included a concise conclusion in the manuscript (line 463 to 470).

In summation, this work proposes an Artificial Neural Network Classifier; AUC (88% for antenatal depression and 85% for antenatal anxiety) and Support Vector Machine Classifier (80% for antenatal depression and 77% for antenatal anxiety) to predict the risk in expecting mothers. Clinical decision support systems are needed and are predicted to be more effective owing to their ability to maintain a consistent quality which is distinctly lacking in low resource settings. The need for high value care with minimal cost while maintaining the standards calls for clinical decision support systems that can intelligently channel our resources to people in low resource settings. (68)

Reviewer 2 Report

This project proposes a machine learning system based on SVM and mult-layer perceptrons for identifying depression and anxiety, which is
an important healthcare concern. This is an extremely important problem,
and describing it as a computational problem will attract attention of
researchers to propose better methods to identify cases. 

The manuscript is well-written and I recommend acceptance, but had a few 
minor comments. First, the terms "antenatal" and "perinatal" were unclear,a
and need to be defined. What type of depression is this study targeted at?
Second, identifying references by "Reference (X)" should be replaced by clickable links. Third, I would like to have seen additional quantitative analysis.
Finally, I would strongly recommend the authors release their anonymized data and code for research purposes.

Author Response

Comment 1

First, the terms "antenatal" and "perinatal" were unclear and need to be defined. What type of depression is this study targeted at?

Response

This study is aimed at antenatal anxiety and depression. We now explain the concepts of antenatal, postnatal, or perinatal in context of mental disorders. We also point out that, lately, the distinction of the time of onset of these disorders have been eliminated. It reads as,

“Depression and anxiety associated with pregnancy has classically been divided into two entities according to time of onset: during pregnancy termed as prenatal or antenatal and post-delivery, termed as postpartum or postnatal. In research and clinical settings, symptoms of pregnancy associated anxiety and depression are either assessed as a continuum of severity scores assessed using psychometric scales such as the Edinburgh Postnatal Depression Scale or as clinician administered diagnoses based on diagnostic criteria of International Statistical Classification of Diseases and Related Health Problems (ICD) or the Diagnostic and Statistical Manual of Mental Disorders (DSM). However, the last version of the DSM (fifth version), the distinction between antenatal or postnatal episodes have been removed. These are now known as "depressive disorder with peripartum/perinatal onset", starting anytime during pregnancy or within the four weeks following delivery. Despite being a major health concern, symptoms of depression during pregnancy – anhedonia, changes in energy, sleep and appetite may be misinterpreted as normal experiences of pregnancy (4).” (Line 72 to 81)

Comment 2

Second, identifying references by "Reference (X)" should be replaced by clickable links.

Response

The references should be replaced with clickable links by the MDPI team hopefully, during formatting.

Comment 3

Third, I would like to have seen additional quantitative analysis.

Response

We have now given a new supplementary table 1 presenting characteristics of study participants. We felt the manuscript was already overloaded with a lot of analyses, and therefore, did not want to overwhelm the reader. So, we have kept several figures as supplementary files as well.

Comment 4

Finally, I would strongly recommend the authors release their anonymized data and code for research purposes.

Response

The dataset associated with this manuscript is available as supplementary file “S1_Data.csv” and the code at github (https://github.com/FJaved1/antenatal-depression-prediction.git.)

This has been mentioned in the manuscript as well.
